# SUPPORT-GUIDED ADVERSARIAL IMITATION LEARNING

## ABSTRACT

We propose Support-guided Adversarial Imitation Learning (SAIL), a generic imitation learning framework that unifies support estimation of the expert policy with the family of Adversarial Imitation Learning (AIL) algorithms. SAIL addresses two important challenges of AIL, including the implicit reward bias and potential training instability. We also show that SAIL is at least as efficient as standard AIL. In an extensive evaluation, we demonstrate that the proposed method mitigates the reward bias and achieves better performance and training stability than other baseline methods on a wide range of benchmark control tasks.

## 1 INTRODUCTION

The class of Adversarial Imitation Learning (AIL) algorithms learns robust policies that imitate an expert's actions from a small number of expert trajectories, without further access to the expert or environment signals. AIL iterates between refining a reward via adversarial training, and reinforcement learning (RL) with the learned adversarial reward. For instance, Generative Adversarial Imitation Learning (GAIL) (Ho & Ermon, 2016) shows the equivalence between some settings of inverse reinforcement learning and Generative Adversarial Networks (GANs) (Goodfellow et al., 2014), and recasts imitation learning as distribution matching between the expert and the RL agent. Similarly, Adversarial Inverse Reinforcement Learning (AIRL) (Fu et al., 2017) modifies the GAIL discriminator to learn a reward function robust to changes in dynamics or environment properties.

AIL mitigates the issue of distributional drift from behavioral cloning (Ross et al., 2011), a classical imitation learning algorithm, and demonstrates good performance with only a small number of expert demonstrations. However, AIL has several important challenges, including implicit reward bias (Kostrikov et al., 2019), potential training instability (Salimans et al., 2016; Brock et al., 2018), and potential sample inefficiency with respect to environment interaction (Sasaki et al., 2019). In this paper, we propose a principled approach towards addressing these issues.

Wang et al. (2019) demonstrated that imitation learning is also feasible by constructing a fixed reward function via estimating the support of the expert policy. Since support estimation only requires expert demonstrations, the method sidesteps the training instability associated with adversarial training. However, we show in Section 4.2 that the reward learned via support estimation deteriorates when expert data is sparse, and leads to poor policy performances.

Support estimation and adversarial reward represent two different yet complementary RL signals for imitation learning, both learnable from expert demonstrations. We unify both signals into *Support-guided Adversarial Imitation Learning* (SAIL), a generic imitation learning framework. SAIL leverages the adversarial reward to guide policy exploration and constrains the policy search to the estimated support of the expert policy. It is compatible with existing AIL algorithms, such as GAIL and AIRL. We also show that SAIL is at least as efficient as standard AIL. In an extensive evaluation, we demonstrate that SAIL mitigates the implicit reward bias and achieves better performance and training stability against baseline methods over a series of benchmark control tasks.

## 2 BACKGROUND

We briefly review the Markov Decision Process (MDP), the context of our imitation learning task, followed by related works on imitation learning.

**Markov Decision Process** We consider an infinite-horizon discounted MDP $(S, A, P, r, p_0, \gamma)$, where $S$ is the set of states, $A$ the set of actions, $P : S \times A \times S \to [0, 1]$ the transition probability, $r : S \times A \to \mathbb{R}$ the reward function, $p_0 : S \to [0, 1]$ the distribution over initial states, and $\gamma \in (0, 1)$ the discount factor. Let $\pi$ be a stochastic policy $\pi : S \times A \to [0, 1]$ with expected discounted reward $\mathbb{E}_\pi(r(s, a)) \triangleq \mathbb{E}(\sum_{t=0}^{\infty} \gamma^t r(s_t, a_t))$ where $s_0 \sim p_0$, $a_t \sim \pi(\cdot|s_t)$, and $s_{t+1} \sim P(\cdot|s_t, a_t)$ for $t \geq 0$. We denote $\pi_E$ the expert policy.

**Behavioral Cloning (BC)** learns a policy $\pi : S \to A$ directly from expert trajectories via supervised learning. BC is simple to implement, and effective when expert data is abundant. However, BC is prone to distributional drift: the state distribution of expert demonstrations deviates from that of the agent policy, due to accumulation of small mistakes during policy execution. Distributional drift may lead to catastrophic errors (Ross et al., 2011). While several methods address the issue (Ross & Bagnell, 2010; Sun et al., 2017), they often assume further access to the expert during training.

**Inverse Reinforcement Learning (IRL)** first estimates a reward from expert demonstrations, followed by RL using the estimated reward (Ng & Russell, 2000; Abbeel & Ng, 2004). Building upon a maximum entropy formulation of IRL (Ziebart et al., 2008), Finn et al. (2016) and Fu et al. (2017) explore adversarial IRL and its connection to Generative Adversarial Imitation Learning (Ho & Ermon, 2016).

**Imitation Learning via Distribution Matching** Generative Adversarial Imitation Learning (GAIL) (Ho & Ermon, 2016) frames imitation learning as distribution matching between the expert and the RL agent. The authors show the connection between IRL and GANs. Specifically, GAIL imitates the expert by formulating a minimax game:

$$\min_\pi \max_{D \in (0,1)} \mathbb{E}_\pi(\log D(s, a)) + \mathbb{E}_{\pi_E}(\log(1 - D(s, a))), \tag{1}$$

where the expectations $\mathbb{E}_\pi$ and $\mathbb{E}_{\pi_E}$ denote the joint distributions over state-actions of the RL agent and the expert, respectively. GAIL is able to achieve expert performance with a small number of expert trajectories on various benchmark tasks. However, GAIL is relatively sample inefficient with respect to environment interaction, and inherits issues associated with adversarial learning, such as vanishing gradients, training instability and overfitting to expert demonstrations (Arjovsky & Bottou, 2017; Brock et al., 2018).

Recent works have improved the sample efficiency and stability of GAIL. For instance, Generative Moment Matching Imitation Learning (Kim & Park, 2018) replaces the adversarial reward with a non-parametric maximum mean discrepancy estimator to sidestep adversarial learning. Baram et al. (2017) improve sample efficiency with a model-based RL algorithm. Kostrikov et al. (2019) and Sasaki et al. (2019) demonstrate significant gain in sample efficiency with off-policy RL algorithms. In addition, Generative Predecessor Models for Imitation Learning (Schroecker et al., 2019) imitates the expert policy using generative models to reason about alternative histories of demonstrated states.

Our proposed method is closely related to the broad family of AIL algorithms including GAIL and adversarial IRL. It is also complementary to many techniques for improving the algorithmic efficiency and stability, as discussed above. In particular, we focus on improving the quality of the learned reward by constraining adversarial reward to the estimated support of the expert policy.

**Imitation Learning via Support Estimation** Alternative to AIL, Wang et al. (2019) demonstrate the feasibility of using a fixed RL reward via estimating the support of the expert policy from expert demonstrations. Connecting kernel-based support estimation (De Vito et al., 2014) to Random Network Distillation (Burda et al., 2018), the authors propose Random Expert Distillation (RED) to learn a reward function based on support estimation. Specifically, RED learns the reward parameter $\hat{\theta}$ by minimizing:

$$\min_{\hat{\theta}} \mathbb{E}_{s,a \sim \pi_E} ||f_{\hat{\theta}}(s, a) - f_\theta(s, a)||_2^2, \tag{2}$$

where $f_\theta : S \times A \to \mathbb{R}^K$ projects $(s, a)$ from expert demonstrations to some embedding of size $K$, with randomly initialized $\theta$. The reward is then defined as:

$$r_{red}(s, a) = \exp(-\sigma||f_{\hat{\theta}}(s, a) - f_\theta(s, a)||_2^2), \tag{3}$$

where $\sigma$ is a hyperparameter. As optimizing Eq. (2) only requires expert data, RED sidesteps adversarial learning, and casts imitation learning as a standard RL task using the learned reward. While RED works well given sufficient expert data, we show in the experiments that its performance suffers in the more challenging setting of sparse expert data.

## 3 METHOD

Formally, we consider the task of learning a reward function $\hat{r}(s, a)$ from a finite set of trajectories $\{\tau_i\}_{i=1}^N$, sampled from the expert policy $\pi_E$ within a MDP. Each trajectory is a sequence of state-action tuples in the form of $\tau_i = \{s_1, a_1, s_2, a_2, ..., s_T, a_T\}$. Assuming that the expert trajectories are consistent with some latent reward function $r^*(s, a)$, we aim to learn a policy that achieves good performance with respect to $r^*(s, a)$ by applying RL on the learned reward function $\hat{r}(s, a)$.

In this section, we first discuss the advantages and shortcomings of AIL to motivate our method. We then introduce Support-guided Adversarial Learning (SAIL), and present a theoretical analysis that compares SAIL with the existing methods, specifically GAIL.

### 3.1 ADVERSARIAL IMITATION LEARNING

A clear advantage of AIL resides in its low sample complexity with respect to expert data. For instance, GAIL requires as little as 200 state-action tuples from the expert to achieve imitation. The reason is that the adversarial reward may be interpreted as an effective exploration mechanism for the RL agent. To see this, consider the learned reward function under the optimality assumption. With the optimal discriminator to Eq. (1) $D^*(s, a) = \frac{p_\pi(s,a)}{p_{\pi_E}(s,a)+p_\pi(s,a)}$, a common reward for GAIL is

$$r_{gail}(s, a) = -\log(D^*(s, a)) = \log\left(1 + \frac{p_{\pi_E}(s,a)}{p_\pi(s,a)}\right) = \log(1 + \phi(s, a)). \qquad (4)$$

Eq. (4) shows that the adversarial reward only depends on the ratio $\phi(s, a) = \frac{p_{\pi_E}(s,a)}{p_\pi(s,a)}$. Intuitively, $r_{gail}$ incentivizes the RL agent towards under-visited state-actions, where $\phi(s, a) > 1$, and away from over-visited state-actions, where $\phi(s, a) < 1$. When $\pi_E$ and $\pi$ match exactly, $r_{gail}$ converges to an indicator function for the support of $\pi_E$, since $\phi(s, a) = 1 \; \forall \; (s, a) \in \text{supp}(\pi_E)$ (Goodfellow et al., 2014). In practice, the adversarial reward is unlikely to converge, as $p_{\pi_E}$ is estimated from a finite set of expert demonstrations. Instead, the adversarial reward continuously drives the agent to explore by evolving the reward landscape.

However, AIL also presents several challenges. Kostrikov et al. (2019) demonstrated that the reward $-\log D(s, a)$ suffers from an implicit survival bias, as the non-negative reward may lead to sub-optimal behaviors in goal-oriented tasks where the agent learns to move around the goal to accumulate rewards, instead of completing the tasks. While the authors resolve the issue by introducing absorbing states, the solution assumes extra RL signals from the environment, including access to the time limit of an environment to detect early termination of training episodes. In Section 4.1, we empirically demonstrate the survival bias on Lunar Lander, a common RL benchmark, by showing that agents trained with GAIL often hover over the goal location[1]. We also show that our proposed method is able to robustly imitate the expert.

Another challenge with AIL is potential training instability. Wang et al. (2019) demonstrated empirically that the adversarial reward could be unreliable in regions where the expert data is sparse, causing the agent to diverge from the intended behavior. When the agent policy is substantially different from the expert policy, the discriminator could differentiate them with high confidence, resulting in very low rewards and significant slow down in training, similar to the vanishing gradient problem in GAN training (Arjovsky & Bottou, 2017).

### 3.2 SUPPORT-GUIDED ADVERSARIAL IMITATION LEARNING

We propose a novel reward function by combining the standard adversarial reward $r_{gail}$ with the corresponding support guidance $r_{red}$.

$$r_{sail}(s, a) = r_{red}(s, a) \cdot r_{gail}(s, a). \qquad (5)$$

SAIL is designed to leverage the exploration mechanism offered by the adversarial reward, and to constrain the agent to the estimated support of the expert policy. Despite being a simple modification, support guidance provides strong reward shaping to address the challenges discussed in the previous

---

[1]The agents still learn to land for some initial conditions.

---

**Algorithm 1** SUPPORT-GUIDED ADVERSARIAL IMITATION LEARNING

---

1: **Input:** Expert trajectories $\tau_E = \{(s_i, a_i)\}_{i=1}^N$, $\Theta$ function models, initial policy $\pi_{\omega_0}$, initial discriminator parameters $w_0$, learning rate $l_D$.

2: $r_{red} = \text{RED}(\Theta, \tau_E)$
3: **for** $i = 0, 1, \ldots$
4:      sample a trajectory $\tau_i \sim \pi$
5:      $w_{i+1} = w_i + l_D \left( \hat{\mathbb{E}}_{\tau_i} (\nabla \log D_{w_i}(s, a)) + \hat{\mathbb{E}}_{\tau_E} (\nabla \log(1 - D_{w_i}(s, a))) \right)$
6:      $r_{gail} : (s, a) \mapsto 1 - D_{w_{i+1}}(s, a)$
7:      $\pi_{\omega_{i+1}} = \text{TRPO}(r_{red} \cdot r_{gail}, \pi_{\omega_i})$.

8: **def** $\text{RED}(\Theta, \tau)$
9:      Sample $\theta \in \Theta$
10:      $\hat{\theta} = \text{MINIMIZE}(f_{\hat{\theta}}, f_\theta, \tau)$
11:      **return** $r_{red} : (s, a) \mapsto \exp(-\sigma || f_{\hat{\theta}}(s, a) - f_\theta(s, a) ||_2^2)$

---

section. As both support guidance and adversarial reward are learnable from expert demonstrations, our method requires no further assumptions that standard AIL.

SAIL addresses the survival bias in goal-oriented tasks by encouraging the agent to stop at the goal and complete the task. In particular, $r_{red}$ shapes the adversarial reward by favoring stopping at the goal against all other actions, as stopping at the goal is on the support of the expert policy, while other actions are not. We demonstrate empirically that SAIL assigns significantly higher reward towards completing the task and corrects for the bias in Section 4.1. To improve training stability, SAIL constrains the RL agent to the estimated support of the expert policy, where $r_{gail}$ provides a more reliable RL signal (Wang et al., 2019). As $r_{red}$ tends to be very small (ideally zero) for $(s, a) \notin \text{supp}(\pi_E)$, $r_{sail}$ discourages the agent from exploring those state-actions by masking away the rewards. This is a desirable property as the quality of the RL signals beyond the support of the expert policy can't be guaranteed. We demonstrate in Section 4.2 the improved training stability on the Mujoco benchmark tasks .

We provide the pseudocode implementation of SAIL in Algorithm 1. The algorithm computes $r_{red}$ by estimating the support of the expert policy, followed by iterative updates of the policy and $r_{gail}$. We apply the Trust Region Policy Optimization (TRPO) algorithm (Schulman et al., 2015) with the reward $r_{sail}$ for policy updates.

**Reward Variants** In practice, we observe that constraining the range of the adversarial reward generally produces lower-variance policies. Specifically, we transform $r_{gail}$ in Eq. (5) from $-\log D(s, a) \in [0, \infty]$ to $1 - D(s, a) \in [0, 1]$. For ease of notation, we refer to the bounded variant as SAIL-b, and the unbounded variant as SAIL. Similarly, we denote the bounded GAIL reward as GAIL-b. We include the comparison between the reward variants in the experiments.

### 3.3 COMPARING SAIL WITH GAIL

In this section, we show that SAIL is at least as efficient as GAIL in its sample complexity for expert data, and provide comparable RL signals on the expert policy's support. We note that our analysis could be similarly applied to other AIL methods, suggesting the broad applicability of our approach.

We begin from the asymptotic setting, where the number of expert trajectories tends to infinity. In this case, both GAIL's, RED's and SAIL's discriminators ultimately recover the expert policy's support at convergence (see Ho & Ermon (2016) for GAIL and Wang et al. (2019) for RED; SAIL follows from their combination). Moreover, for both GAIL and SAIL, the expert and agent policy distributions match exactly at convergence, implying a successful imitation learning. Therefore, it is critical to characterize the rates of convergence of the two methods, namely their relative sample complexity with respect to the number of expert demonstrations.

Formally, let $(s, a) \notin \text{supp}(\pi_E)$. Prototypical learning bounds for an estimator of the support $\hat{r} \geq 0$ provide high probability bounds in the form of $\mathbb{P}(\hat{r}(s, a) \leq c \log(1/\delta)n^{-\alpha}) > 1 - \delta$ for

any confidence $\delta \in (0, 1]$, with $c$ a constant not depending on $\delta$ or the number $n$ of samples (i.e., expert state-actions). Here, $\alpha > 0$ represents the learning rate, namely how fast the estimator is converging to the support. By choosing the reward in Eq. (5), we are leveraging the faster learning rates between $\alpha_{red}$ and $\alpha_{gail}$, with respect to support estimation. At the time being, no results are available to characterize the sample complexity of GAIL (loosely speaking, the $\alpha$ and $c$ introduced above). Therefore, we proceed by focusing on a relative comparison with SAIL. In particular, we show the following (see appendix for a proof).

**Proposition 1.** *Assume that for any* $(s, a) \notin \text{supp}(\pi_E)$ *the rewards for RED and GAIL have the following learning rates in estimating the support*

$$\mathbb{P}\left(r_{red}(s, a) > \frac{c_{red} \log \frac{1}{\delta}}{n^{\alpha_{red}}}\right) \leq \delta \qquad \mathbb{P}\left(r_{gail}(s, a) > \frac{c_{gail} \log \frac{1}{\delta}}{n^{\alpha_{gail}}}\right) \leq \delta. \tag{6}$$

*Then, for any* $\delta \in (0, 1]$ *and any* $(s, a) \notin \text{supp}(\pi_E)$*, the following holds*

$$r_{sail}(s, a) \leq \min\left(\frac{c_{red} R_{gail}}{n^{\alpha_{red}}}, \frac{c_{gail} R_{red}}{n^{\alpha_{gail}}}\right) \log \frac{1}{\delta}, \tag{7}$$

*with probability at least* $1 - \delta$*, where* $R_{red}$ *and* $R_{gail}$ *are the upper bounds for* $r_{red}$ *and* $r_{gail}$*, respectively.*

Eq. (7) shows that SAIL is at least as fast as the faster among RED and GAIL with respect to support estimation, implying that SAIL is at least as efficient as GAIL in the sample complexity for expert data. Eq. (7) also indicates the quality of the learned reward, as state-actions outside the expert's support should be assigned minimum reward.

**Proposition 2.** *For any* $(s, a) \in \text{supp}(\pi_E)$ *and any* $\delta \in (0, 1]$*, we assume that*

$$\mathbb{P}\left(|r_{red}(s, a) - 1| > \frac{c_{red} \log \frac{1}{\delta}}{n^{\alpha_{red}}}\right) < \delta. \tag{8}$$

*The following event holds with probability at least* $1 - \delta$ *that*

$$|r_{sail}(s, a) - r_{gail}(s, a)| \leq \frac{c_{red} R_{gail}}{n^{\alpha_{red}}} \log \frac{1}{\delta}. \tag{9}$$

Eq. (9) shows that on the expert policy's support, $r_{sail}$ is close to $r_{gail}$ up to a precision that improves with the number of expert state-actions. SAIL thus provides RL signals comparable to GAIL on the expert policy's support.

It is also worth noting that the analysis could explain why $r_{red} + r_{gail}$ is a less viable approach for combining the two RL signals. The analogous bound to Eq. (7) would be the sum of errors from the two methods, implying the *slower* of the two learning rates, while Eq. (9) would improve only by a constant, as $R_{gail}$ would be absent from Eq. (9). Our preliminary experiments indicated that $r_{red} + r_{gail}$ performed noticeably worse than Eq. (5).

Lastly, we comment on whether the assumptions in Eqs. (6) and (8) are satisfied in practice. Following the kernel-based version of RED (Wang et al., 2019), we can borrow previous results from the set learning literature, which guarantee RED to have a rate of $\alpha_{red} = 1/2$ (De Vito et al., 2014; Rudi et al., 2017). These rates have been shown to be *optimal*. Any estimator of the support cannot have faster rates than $n^{-1/2}$, unless additional assumptions are imposed. Learning rates for distribution matching with GANs are still an active area of research, and conclusive results characterizing the convergence rates of these estimators are not available. We refer to Singh et al. (2018) for an in-depth analysis of the topic.

## 4 EXPERIMENTS

We evaluate the proposed method against BC, GAIL and RED on Lunar Lander and six Mujoco control tasks including Hopper, Reacher, HalfCheetah, Walker2d, Ant, and Humanoid. We omit evaluation against methods using off-policy RL algorithms, as they are not the focus of this work. We also note that support guidance is complementary to such methods.

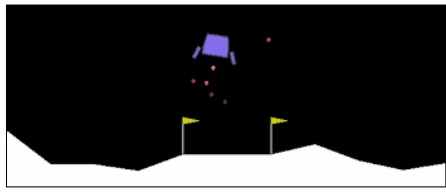

|        | Default            | No-terminal        |
|--------|--------------------|--------------------|
| BC     | 100.38 ± 130.91    | 100.38 ± 130.91    |
| RED    | 13.75 ± 53.43      | -39.33 ± 24.39     |
| GAIL   | 258.30 ± 28.98     | 169.73 ± 80.84     |
| GAIL-b | 250.53 ± 67.07     | -69.33 ± 79.76     |
| SAIL   | 257.02 ± 20.66     | 237.96 ± 49.70     |
| SAIL-b | **262.97 ± 18.11** | **256.83 ± 20.99** |
| Expert | 253.58 ± 31.27     | 253.58 ± 31.27     |

Figure 1: The task of Lunar Lander requires landing the spacecraft between the flags without crashing.

Table 1: Average environment reward and standard deviation on Lunar Lander, evaluated over 50 runs for the default and no-terminal environment.

### 4.1 LUNAR LANDER

We demonstrate that SAIL variants mitigate the survival bias in Lunar Lander (Fig. 1) from OpenAI Gym (Brockman et al., 2016), while other baseline methods imitate the expert inconsistently. In this task, the agent is required to control a spacecraft to safely land between the flags. A human expert provided 10 demonstrations for this task as an imitation target.

We observe that even without the environment reward, Lunar Lander provides a natural RL signal by terminating episodes early when crashes are detected, thus encouraging the agent to avoid crashing. Consequently, all methods are able to successfully imitate the expert and land the spacecraft appropriately. SAIL variants perform slightly better than GAIL variants on the average reward, and achieve noticeably lower standard deviation. The average performances and the standard deviations evaluated over 50 runs are presented in Table 1.

To construct a more challenging task, we disable all early termination feature of the environment, thus removing the environment RL signals. In this no-terminal environment, a training episode only ends after the time limit. We present each algorithm's performance for the no-terminal setting in Table 1. SAIL variants outperform GAIL variants. Specifically, we observe that GAIL learns to land for some initial conditions, while exhibit survival bias in other scenarios by hovering at the goal. In contrast, SAIL variants are still able to recover the expert policy.[2]

To visualize the shaping effect from support guidance, we plot the average learned reward for GAIL, SAIL-b and RED at goal states. The goal states are selected from the expert trajectories and satisfy two conditions: 1) touching the ground (the state vector has indicator variables for ground contact), and 2) has "no op" as the corresponding action. As the adversarial reward functions are dynamic, we snapshot the learned rewards when the algorithms obtain their best policies, respectively. Fig. 3 shows the average rewards for each available action, averaged across all the goal states. Compared against the other algorithms, SAIL-b assigns a significantly higher reward to "no op", which facilitates the agent learning. Though GAIL and RED still favor "no op" to other actions, the differences in reward are much smaller, causing less consistent landing behaviors.

We further observe that all evaluated AIL methods oscillate between partially hovering behavior and landing behavior during policy learning. The observation suggests that our method only partially addresses the survival bias, a limitation we will tackle in future works. This is likely caused by SAIL's non-negative reward, despite the beneficial shaping effect from support estimation. For additional experiment results and discussion on Lunar Lander, please refer to the appendix.

### 4.2 MUJOCO TASKS

Mujoco control tasks have been commonly used as the standard benchmark for AIL. We evaluate SAIL against GAIL, RED and BC on Hopper, Reacher, HalfCheetah, Walker2d, Ant and Humanoid. We adopt the same experimental setup presented in Ho & Ermon (2016) by sub-sampling the expert trajectories every 20 samples. Consistent with the observation from Kostrikov et al. (2019), our preliminary experiments show that sub-sampling presents a more challenging setting, as BC is competitive with AIL when full trajectories are used. In our experiments, we also adopt the minimum

---

[2]A illustrative video is available at `https://vimeo.com/361835881`

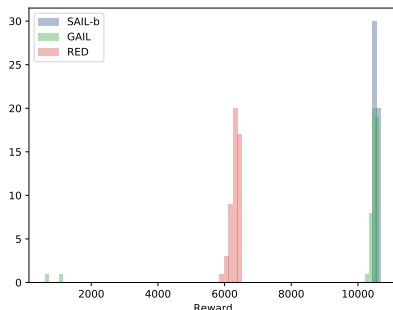 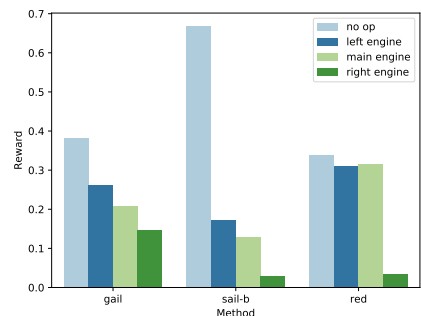

Figure 2: Performance histogram of 50 evaluation runs on Humanoid for RED, GAIL, and SAIL-b. SAIL-b imitates the expert consistently. GAIL has undesirable failure cases, with rewards of less than 1000 (bottom left corner). RED is consistent though sub-optimal.

Figure 3: Average reward assignment at the goal states by different algorithms. SAIL-b assigns significantly higher reward to "no op", enabling the agent to learn the appropriate landing behaviors. Other algorithms fail to imitate the expert consistently.

|        | Hopper           | Reacher         | Cheetah            | Walker             | Ant                 | Humanoid             |
|--------|------------------|-----------------|--------------------|--------------------|---------------------|----------------------|
| BC     | $312.3 \pm 34.5$ | $-8.8 \pm 3.3$  | $1892.0 \pm 206.9$ | $248.2 \pm 117.8$  | $1752.0 \pm 434.8$  | $539.4 \pm 185.7$    |
| RED    | $1056.5 \pm 0.5$ | $-9.1 \pm 4.1$  | $-0.2 \pm 0.7$     | $2372.8 \pm 8.8$   | $1005.5 \pm 8.6$    | $6012.0 \pm 434.9$   |
| GAIL   | $\mathbf{3826.5} \pm \mathbf{3.2}$ | $-9.1 \pm 4.4$ | $4604.7 \pm 77.6$  | $5295.4 \pm 44.1$  | $1013.3 \pm 16.0$   | $8781.2 \pm 3112.6$  |
| GAIL-b | $3810.5 \pm 8.1$ | $-8.3 \pm 2.5$  | $4510.0 \pm 68.0$  | $5388.1 \pm 161.2$ | $3413.1 \pm 744.7$  | $10132.5 \pm 1859.3$ |
| SAIL   | $3824.7 \pm 6.6$ | $-7.5 \pm 2.7$  | $\mathbf{4747.5} \pm \mathbf{43.4}$ | $5293.0 \pm 590.9$ | $3330.4 \pm 729.4$  | $9292.8 \pm 3190.0$  |
| SAIL-b | $3811.6 \pm 3.8$ | $\mathbf{-7.4} \pm \mathbf{2.5}$ | $4632.2 \pm 59.1$ | $\mathbf{5438.6} \pm \mathbf{18.4}$ | $\mathbf{4176.3} \pm \mathbf{203.1}$ | $\mathbf{10589.6} \pm \mathbf{52.2}$ |

Table 2: Episodic reward and standard deviation on the Mujoco tasks by different methods evaluated over 50 runs. SAIL-b achieves overall the best performance, with significantly lower standard deviation, indicating the robustness of the learned policies.

number of expert trajectories specified in Ho & Ermon (2016) for each task. More details on experiment setup are available in the appendix.

We apply each algorithm using 5 different random seeds in all Mujoco tasks. Table 2 shows the performance comparison between the evaluated algorithms. We report the mean performance and standard deviation for each algorithm over 50 evaluation runs, choosing the best policies obtained for each algorithm out of the 5 random seeds.

The results show that SAIL-b is comparable to GAIL on Hopper, and outperform the other methods on all other tasks. We note that RED significantly underperforms in the sub-sampling setting, while Wang et al. (2019) used full trajectories in their experiments. Across all tasks, SAIL-b generally achieves lower standard deviation compared to other algorithms, in particular for Humanoid, indicating the robustness of the learned policies.

We stress that standard deviation is also a critical metric, as it indicates the robustness of the learned policies when presented with different states. For instance, the large standard deviations in Humanoid are caused by occasional crashes, which may be highly undesirable depending on the intended applications. To illustrate robustness of the learned policies, we plot the histogram of all 50 evaluations in Humanoid for RED, GAIL-b and SAIL-b in Fig. 2. The figure shows that SAIL-b performs consistently with expert performance. Though GAIL-b appears to be only slightly worse in average performance, the degradation is caused by occasional and highly undesirable crashes, suggesting incomplete imitation of the expert. RED performs the worst in average performance, but is consistent with no failure modes detected. The result suggests that the proposed method combines the advantages of both support guidance and adversarial learning.

Comparing SAIL against SAIL-b, we observe that the bounded variant generally produces policies with smaller standard deviations and better performances, especially for Ant and Humanoid. This

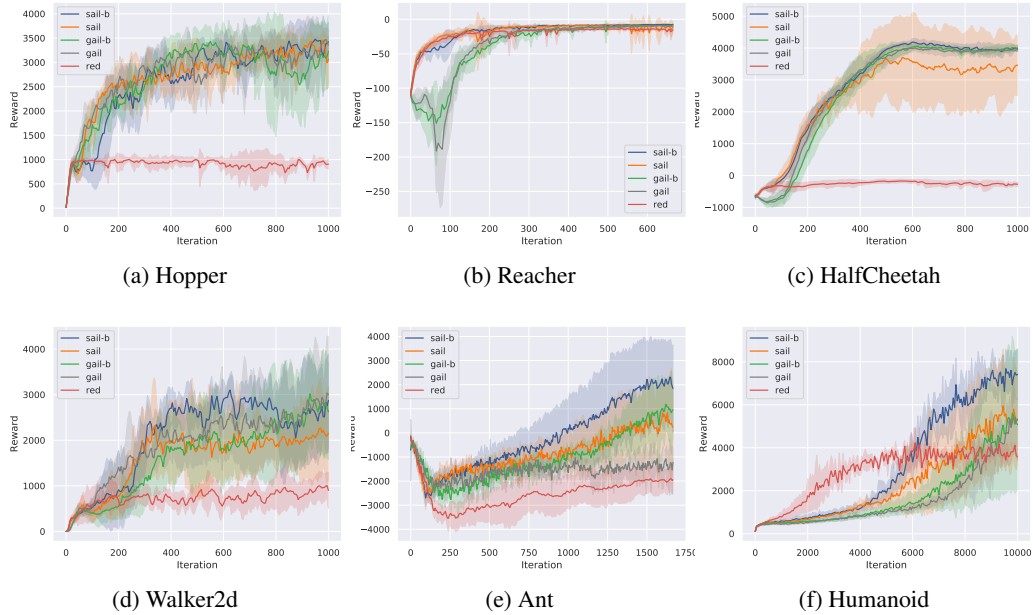

Figure 4: Training progress for RED, GAIL, GAIL-b, SAIL, and SAIL-b. Consistent with our theoretical analysis, SAIL-b (blue) is more stable and sample efficient in Reacher, Ant and Humanoid, and comparable to other algorithms for the remaining tasks.

is likely due to the fact that SAIL-b receives equal contribution from both support guidance and adversarial learning, as $r_{red}$ and $r_{gail}$ have the same range in this formulation. In addition, we note that GAIL fails to imitate the expert in Ant, while GAIL-b performs significantly better. The results suggest that restricting the range of the adversarial reward could improve performance.

## 4.3 TRAINING STABILITY AND SAMPLE EFFICIENCY

To assess the sensitivity with respect to random seeds, we plot the training progress against number of iterations for the evaluated algorithms in Fig. 4, Each iteration consists of 1000 environment steps. The figure reports mean and standard deviation of each algorithm, across the 5 random seeds.

Fig. 4 shows that SAIL-b is more sample efficient and stable in Reacher, Ant and Humanoid tasks; and is comparable to the other algorithms in the remaining tasks. Consistent with our analysis in Section 3.3, SAIL-b appears at least as efficient as GAIL even when the support guidance (i.e., the performance of RED) suffers from insufficient expert data in Hopper, HalfCheetah and Walker2d. In Reacher, Ant and Humanoid, SAIL-b benefits from the support guidance and achieves better performance and training stability. In particular, we note that without support guidance, GAIL fails to imitate the expert in Ant (Fig. 4e). Similar failures were also observed in Kostrikov et al. (2019). GAIL is also more sensitive to initial conditions: in Humanoid, GAIL converged to sub-optimal policies in 2 out 5 seeds. Lastly, while RED improves noticeably faster during early training in Humanoid, it converged to a sub-optimal policy eventually.

## 5 CONCLUSION

In this paper, we propose Support-guided Adversarial Imitation Learning by combining support guidance with adversarial imitation learning. Our approach is complementary to existing adversarial imitation learning algorithms, and addresses several challenges associated with them. More broadly, our results show that expert demonstrations contain rich sources of information for imitation learning. Effectively combining different sources of reinforcement learning signals from the expert demonstrations produces more efficient and stable algorithms by constraining the policy search space; and appears to be a promising direction for future research.

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

## A  PROOF FOR PROPOSITION 1 AND 2

Observe that for any $(s, a) \in S \times A$

$$r_{sail}(s, a) = r_{red}(s, a) \cdot r_{gail}(s, a) \leq \min(r_{red}(s, a) R_{gail}, r_{gail}(s, a) R_{red}). \qquad (10)$$

By the assumption on the learning rate in Eq. (6), one of the two following events holds with probability at least $1 - \delta$, for any $(s, a) \notin \text{supp}(\pi_E)$ and $\delta \in (0, 1]$

$$r_{red}(s, a) \leq \frac{c_{red} \log \frac{1}{\delta}}{n^{\alpha_{red}}} \qquad \text{or} \qquad r_{gail}(s, a) \leq \frac{c_{gail} \log \frac{1}{\delta}}{n^{\alpha_{gail}}}. \qquad (11)$$

Plugging the above upper bounds into Eq. (10) yields the desired result in Eq. (7).

By assumption in Eq. (8) the following event holds with probability at least $1 - \delta$ for $(s, a) \in \text{supp}(\pi_E)$.

$$|r_{red}(s, a) - 1| \leq \frac{c_{red} \log \frac{1}{\delta}}{n^{\alpha_{red}}}. \qquad (12)$$

Plugging this inequality in the definition of $r_{sail}$, we obtain

$$|r_{sail}(s, a) - r_{gail}(s, a)| = |r_{gail}(s, a)(r_{red}(s, a) - 1)| \qquad (13)$$

$$\leq |r_{gail}(s, a)||r_{red}(s, a) - 1| \qquad (14)$$

$$\leq \frac{c_{red} R_{gail} \log \frac{1}{\delta}}{n^{\alpha_{red}}}, \qquad (15)$$

$$\square$$

## B  EXPERIMENT DETAILS

The experiments are based on OpenAI's baselines[3] and the original implementation of RED[4]. We adapted the code from RED[4] for our experiments, and used the accompanying dataset of expert trajectories. 4 Nvidia GTX1070 GPUs were used in the experiments.

Table 3 shows the environment information, number of environment steps and number of expert trajectories used for each task. Each full trajectory consists of 1000 $(s, a)$ pairs. They are sub-sampled during the experiments.

---

[3] https://github.com/openai/baselines
[4] https://github.com/RuohanW/RED

| Task | State Space | Action Space | Trajectories | Env Steps | Exp Performance |
|------|-------------|--------------|--------------|-----------|-----------------|
| Hopper-v2 | 11 | 3 | 4 | $3 \times 10^6$ | $3777.8 \pm 3.8$ |
| Reacher-v2 | 11 | 2 | 4 | $1 \times 10^6$ | $-3.7 \pm 1.4$ |
| HalfCheetah-v2 | 17 | 6 | 4 | $3 \times 10^6$ | $4159.8 \pm 93.1$ |
| Walker2d-v2 | 17 | 6 | 4 | $3 \times 10^6$ | $5505.8 \pm 81.4$ |
| Ant-v2 | 111 | 8 | 4 | $5 \times 10^6$ | $4821.0 \pm 107.4$ |
| Humanoid-v2 | 376 | 17 | 80 | $3 \times 10^7$ | $10413.1 \pm 47.0$ |

Table 3: Environment information, number of expert trajectories and environment steps used for each task

### B.1 NETWORK ARCHITECTURE

The default policy network from OpenAI's baselines are used for all tasks: two fully-connected layers of 100 units each, with tanh nonlinearities. The discriminator networks and the value function networks use the same architecture.

RED and SAIL use RND Burda et al. (2018) for support estimation. We use the default networks from RED[4]. We set $\sigma$ following the heuristic in Wang et al. (2019) that $(s, a)$ from the expert trajectories mostly have reward close to 1.

### B.2 HYPERPARAMETERS

For fair comparisons, all algorithms shared hyperparameters for each task. We present them in the table below, including discriminator learning rate $l_D$, discount factor $\gamma$, number of policy steps per iteration $n_G$, and whether the policy has fixed variance. All other hyperparameters are set to their default values from OpenAI's baselines.

| Task Name | $\gamma$ | $l_D$ | $n_G$ | Fixed Variance |
|-----------|----------|-------|-------|----------------|
| Hopper | 0.99 | 0.0003 | 3 | False |
| Reacher | 0.99 | 0.0003 | 3 | False |
| HalfCheetah | 0.99 | 0.0003 | 3 | False |
| Walker2d | 0.99 | 0.0003 | 3 | False |
| Ant | 0.99 | 0.0001 | 3 | False |
| Humanoid | 0.99 | 0.0001 | 5 | False |

Table 4: Hyperparameters used for each tasks

## C ADDITIONAL RESULTS ON LUNAR LANDER

In the default environment, Lunar Lander contains several terminal states, including crashing, flying out of view, and landing at the goal. In the *no-terminal* environment, all terminal states are disabled, such that the agent must solely rely on the expert demonstrations for training signals.

To compare our method with the technique of introducing virtual absorbing state (AS) (Kostrikov et al., 2019), we also construct a *goal-terminal* environment where the only terminal state is successful landing at the goal, because the AS technique cannot be directly applied in the no-terminal environment. We present the results in Appendix C.

The results suggest that AS overall improves both the mean performance and standard deviations for both GAIL and SAIL. Specifically, the technique is able to mitigates the survival bias in GAIL significantly. However, SAIL still compares favorably to the technique in the goal-terminal environment. Further, since AS and support guidance is not mutually exclusive, we also combine them and report the performances. The results suggest that support guidance is compatible with AS, and achieves overall the best performance with low standard deviations.

|           | Default          | Goal-terminal     | No-terminal       |
|-----------|------------------|-------------------|-------------------|
| GAIL      | $258.30 \pm 28.98$ | $-7.16 \pm 31.64$ | $-69.33 \pm 79.76$ |
| GAIL-b    | $250.53 \pm 67.07$ | $4.16 \pm 107.37$ | $169.73 \pm 80.84$ |
| SAIL      | $257.02 \pm 20.66$ | $261.07 \pm 35.66$ | $237.96 \pm 49.70$ |
| SAIL-b    | $262.97 \pm 18.11$ | $252.07 \pm 67.22$ | $256.83 \pm 20.99$ |
| GAIL + AS   | $271.46 \pm 11.90$ | $110.22 \pm 119.25$ | - |
| GAIL-b + AS | $269.97 \pm 16.48$ | $186.02 \pm 98.27$  | - |
| SAIL + AS   | $274.89 \pm 12.82$ | $254.58 \pm 25.40$  | - |
| SAIL-b + AS | $270.33 \pm 15.86$ | $258.30 \pm 20.75$  | - |

Table 5: Average environment reward and standard deviation on Lunar Lander, evaluated over 50 runs for the default, goal-terminal and no-terminal environment.

The results also suggest that both AS and support guidance partially mitigate the reward bias, but don't fully solve it. We will further explore this issue in future work.

