# OpenReview forum: "Support-guided Adversarial Imitation Learning"
_ICLR.cc/2020/Conference — Reject_

### Official Review · AnonReviewer2 · 2019-10-18
**Official Blind Review #2**

**Rating:** 6

**Review:**

**Summary of the paper:
The paper proposes an IL method named support-guided adversarial IL (SAIL), which is based on generative adversarial IL (GAIL) (Ho and Ermon, 2016) and random expert distillation (RED) (Wang et al., 2019). The key idea of SAIL is to construct a reward function by multiplying reward functions learned by GAIL and RED. This multiplication yields two benefits; 1) it handles the issue of biased reward in GAIL, since state-action pairs outside the expert’s support are assigned low reward values. 2) SAIL’s reward is more reliable than RED’s reward for state-action pairs inside the expert’s support. The authors show that SAIL is at least as fast as than GAIL in terms of the sample complexity. Experiments on continuous control benchmarks show that SAIL is overall more stable than GAIL.

**Rating:
The paper proposes a simple but effective combination of existing methods. The proposed method is well motivated and performs well on benchmarks. Still, the paper has some issues regarding justification, clarity, and evaluation, which should be addressed (see below). I vote for weak acceptance.

**Major comments/questions:
- No guarantee of the optimality of the learned policy.
Can it be guaranteed that SAIL learns the expert policy? (assuming the expert policy is realizable). Propositions 1 and 2 show the convergence of the support estimation, but these results are not related to the optimality of a policy learned with the reward function. This is an important point for justifying SAIL, since SAIL does not perform distribution matching to learn the expert policy, and it also does not perform IRL to learn the reward function. Therefore, SAIL lacks the optimality guarantee from both distribution matching and IRL perspectives. Please address and clarify this point.

- Clarity in the theoretical analysis.
In the theoretical analysis, the paper assumes a rate of GAIL for support estimation. This is quite confusing, since GAIL performs distribution matching and does not estimate the support. Also, given that r_gail = -log D(s,a), the reward’s upper-bound (R_gail) is infinity and the bound in Eq. (9) is not informative.

- The reward r_red is constant at the optimal.
Eq. (2) and Eq. (3) imply that, for state-action pairs from the expert’s state-action distribution, r_red is constant at the optimal. Specifically, the optimal solution of Eq. (2) is \hat{\theta} = \theta, which yields to a constant value of r_red(s,a) in Eq. (3). In this scenario, SAIL is equivalent to GAIL for the expert state-action distribution. This means that Eq. (2) should not be optimized until optimal, and some early stopping criteria are required. Does this scenario (constant value of r_red) occur in the experiments?

- IRL baseline methods.
The paper should compare SAIL to methods which aim to handle the bias in reward function, e.g., DAC (Kostrikov et al. 2019). While DAC requires the time limit, this time limit is known in the benchmark tasks. Also, IRL methods such as AIRL (Fu et al., 2018) should be compared, since IRL methods are better than GAIL at handling bias in reward function (Kostrikov et al. 2019).

**Minor comments/questions:
- Typos: "offline RL algorithms" should be "off-policy RL algorithms". Line 5 of Algorithm 1 should perform gradient ascent instead of gradient descent. An expectation over state-action distribution of expert is missing from Eq. (2).

- What are the bold numbers in table 1 and 2 indicating? Why does the Hopper task have two bold numbers, but the other tasks have only one?

--After author response--
I have read the author response and other reviews. I thank the authors for including additional experiments. However, the authors' arguments regarding optimality do not fully address my comments (see below). I will keep the vote of weak acceptance.

The authors argue that "In this asymptotic case, SAIL is equivalent to performing distribution matching via GAIL with the additional *constraint* that candidate distributions need to have the same support of the expert distribution". However, the support of the expert distribution may coincide with the entire state-action space, which makes the additional constraint uninformative in the asymptotic case. Specifically, the expert distribution coincides with the state-action space when the expert policy has an infinite support (e.g., the expert policy is Gaussian). Assuming the asymptotic case, the support estimation in RED will give an indicator function over an entire state-action space, and the support constraint in SAIL is always satisfied. In other words, SAIL is exactly equivalent to GAIL in this case. For these reasons, the authors' arguments regarding optimality do not fully address my comments. I think additional assumptions are required, e.g., the expert policy needs to have a finite support or be deterministic.




**Experience Assessment:**

I have published one or two papers in this area.

**Review Assessment: Checking Correctness Of Derivations And Theory:**

I assessed the sensibility of the derivations and theory.

**Review Assessment: Checking Correctness Of Experiments:**

I assessed the sensibility of the experiments.

**Review Assessment: Thoroughness In Paper Reading:**

I read the paper at least twice and used my best judgement in assessing the paper.

---

> ### Author Response · Authors · 2019-11-11
> **Response to Reviewer 2**
>
> We thank the reviewer for the constructive feedback.
>
> - Clarification on method justification & theoretical analysis
> As the reviewer pointed out, distribution matching does not necessarily imply support estimation (it depends on the metric used for the distribution matching). However, following the intuition in motivating this work, restricting the distribution matching problem to the support of the target distribution (the expert) might help the overall learning.  In this sense, the results in Prop. 1 and 2 show that, even if GAIL were to perform a very slow support estimation (or no support estimation at all), the distribution matching process would be constrained onto the expert’s support, thanks to RED.
>
> In the asymptotic setting, RED converges to an indicator function on the support of the expert policy: r_red(s, a) =1 if (s, a) belongs to the support of the expert policy, and 0 otherwise.
>
> Therefore, In this asymptotic case, SAIL is equivalent to performing distribution matching via GAIL with the additional *constraint* that candidate distributions need to have the same support of the expert distribution. Since we are restricting candidate estimators to the support of the target distribution, distribution matching methods (e.g. GAIL) can still be adopted.
>
> - Clarification on why optimizing Eq. 2 doesn’t recover the \theta and thus render the red reward a constant
> The technique of random network distillation was first proposed in (Burda et al. 2018), which states that empirically, standard training would converge to a local minimum other than the randomly initialized \theta. We have similar observations in our experiment, and thus no special treatment (e.g. early stopping) is used.
>
> - Clarification on additional comparison
> We have included a comparison with the absorbing state (AS) technique (Kostrikov et al. 2019) in Table 5 from Appendix. We also run SAIL combined with AS, since the two methods are not mutually exclusive.
>
> The results are:
> Default env
> GAIL 258.30±28.98 | GAIL+AS  271.46±11.90 | SAIL 262.97±18.11 | SAIL+AS 270.33±15.86
> Modified env (renamed as “Goal-terminal” env in the appendix)
> GAIL -7.16±31.64 | GAIL+AS 110.22±119.25 | SAIL 252.07±67.22 | SAIL+AS 258.30±20.75
>
> AS improves GAIL significantly in both environments. In particular, in the default environment, GAIL+AS has performance comparable to (or slightly better than) SAIL.
>
> However, we also observe that: 1) SAIL+AS either outperforms (or is comparable to) GAIL+AS, showing that the proposed approach is generally more favorable than GAIL. Also, SAIL+AS has much smaller variance than standard SAIL. 2) In the modified environment, GAIL+AS is unable  to reach the expert’s performance and suffers from a high variance, even when it improves significantly upon GAIL. On the contrary, SAIL and SAIL+AS are on-par with the expert.
>
> To better study the effect of SAIL, we further modified LunarLander to contain no terminal states at all. We refer to this setting as NoTerm. Here each episode ends only after a fixed time limit (1000 steps). In this environment, the absorbing state (AS) method is not applicable. We obtain the following returns (updated Table 1 in the paper):
>
> GAIL 169.73±80.84 | SAIL 256.83±20.99
>
> showing that SAIL is significantly more robust than GAIL also in this setting.
>
> In summary, the results on all variants of LunarLander suggests that AS and SAIL both mitigate the implicit reward bias.
>
> - On minor points
> We thank the reviewer for pointing out the typos, they have been fixed.
> The bold font highlights the best performance of each row for ease of reading. Hopper had 2 because we considered them comparable. We have updated the paper to remove the additional highlighting.

---

### Official Review · AnonReviewer1 · 2019-10-23
**Official Blind Review #1**

**Rating:** 6

**Review:**

The paper proposes an imitation learning algorithm that combines support estimation with adversarial training. The key idea is simple: multiply the reward from Random Expert Distillation (RED) with the reward from Generative Adversarial Imitation Learning (GAIL). The new reward combines the best of both methods. Like the GAIL reward, the new reward encourages exploration and can be estimated from a small number of demonstrations. Like the RED reward, the new reward avoids survival bias and is more stable than the adversarial reward.

I have a concern regarding the Lunar Lander experiment. Were the demonstrations generated in the modified environment? If they were generated in the original environment (with early termination), this may have unintentionally created a state distribution mismatch between the demonstration environment and training environment that unfairly hurts the GAIL baseline's performance. If the demonstrations were instead generated in the modified environment (without early termination) where the agent is actually trained, the demonstrations would contain many self-loop transitions at the goal state, and GAIL would likely not exhibit survival bias.

I am also a bit concerned about the MuJoCo results. The stochasticity of the demonstrations and the evaluation trajectories may have a significant effect on the standard deviation of rewards. Was a stochastic policy or a deterministic policy used to generate the demonstrations? Were the evaluation trajectories generated by rolling out the stochastic imitation policy, or by rolling out a deterministic version of the imitation agent? Also, could the authors provide the mean and standard deviation of rewards in the demonstrations in Tables 1-2 and Figure 4? It would be nice to establish a rough upper bound on the performance of the imitation methods.

Update:
After reading the author response, I have increased my score from 3 to 6.

**Experience Assessment:**

I have read many papers in this area.

**Review Assessment: Checking Correctness Of Derivations And Theory:**

I assessed the sensibility of the derivations and theory.

**Review Assessment: Checking Correctness Of Experiments:**

I assessed the sensibility of the experiments.

**Review Assessment: Thoroughness In Paper Reading:**

I read the paper at least twice and used my best judgement in assessing the paper.

---

> ### Author Response · Authors · 2019-11-11
> **Response to Reviewer 1**
>
> We thank the reviewer for the constructive feedback.
>
> - Clarification on LunarLander experiments
> We first note that the expert demonstrations for LunarLander are generated by a human expert, rather than learned via RL. In the paper, we use the demonstration trajectories generated in the default environment.
>
> To assess the potential impact of distribution mismatch, we have “padded” each expert demonstration with loop transitions at the goal state with “no op” action, and re-run the experiment. This is exactly the behavior adopted by the expert after a successful landing (the expert stops giving commands to the lunar lander once it lands). Interestingly, in this setting the training process of the discriminator for the AIL method becomes highly unstable, as the expert demonstration is now dominated by (s, a) = (goal_state, no_op). As the discriminator is trained stochastically via mini-batches, random sampling could lead to a batch of real data with only (goal_state, no_op) present, which destabilizes training.
>
> - Clarifications on Mujoco results
> Our experiment setup follows OpenAI’s reference implementation: a deterministic policy is used for each task to generate the demonstrations to ensure consistent performance. For reproducibility, The evaluations were generated by rolling out the learned policies deterministically. The expert performance is updated in Table 3 from the appendix, and included below.
>
> Hopper 3777.8±3.8 | Reacher -3.7±1.4 | HalfCheetah 4159.8±93.1 | Walker2d 5505.8±81.4 | Ant 4821.0±107.4 | Humanoid 10413.1±47.
>
> Therefore, the results from our paper suggest that the low variance policies can be attributed to our proposed method.

---

### Official Review · AnonReviewer3 · 2019-10-23
**Official Blind Review #3**

**Rating:** 1

**Review:**

This paper proposes an approach for improving adversarial imitation learning, by combining it with support-estimation-based imitation learning. In particular, the paper explores a combination of GAIL (Ho and Ermon, 2016) and RED (Wang et. al., 2019), where the reward for the policy-gradient is a product of the rewards obtained from them separately. The motivation is that, while AIL methods are sample-efficient (in terms of expert data) and implicitly promote useful exploration, they could be unreliable outside the support of the expert policy. Therefore, augmenting them by constraining the imitator to the support of the expert policy (with a method such as RED) could result in an overall better imitation learning algorithm.

While the motivation and intuition are clear to me, I have reservations about the claims made in the abstract and the experimental sections:

1.	SAIL is an effective method for solving the reward bias in AIL.
The reward in SAIL is “always” non-negative (product of 2 non-negative terms), making the method a very ad-hoc way of getting around the reward bias problem, especially when compared to other methods such as those which estimate the value function of the absorbing state (Kostrikov et. al. 2019). Consider a simple chain MDP with 3 states A, B and a terminal state T. The actions are left/right from each state. Let the expert trajectory be A->B->T. Also, for SAIL, consider perfect support estimation with an optimal RED-network. When at B, the agent can terminate with a right-action and collect some reward. But taking left and collecting 0 reward (due to perfect support estimation) makes it land in A, from where it can now achieve a positive reward for the A->B transition, and repeat the process. Hence, one could always create MDPs where the Q value of B->A is higher than B->T.

	The Lunar-Lander environment (with certain parameters) in Section 4.1 appears to present a scenario where SAIL get arounds the reward bias, but this doesn’t remove my doubts over the generalization of this approach. Also, in Table 1, why does GAIL not hover above the landing spot even in the default case? If the reward bias is strong there, with sufficient exploration, the agent should converge to the same policy as in the modified case.

Figure 3 is concerning for the same reason as above. It shows the immediate reward at the goal state, and points that SAIL has large reward for no-op action. The issue is that RL optimizes for actions that have the maximum Q value, not the action with the maximum immediate reward.


2.	I would recommend that the authors refer to the original GAIL algorithm as “GAIL” in the experiments section, and their practical stabilization trick as “GAIL-bounded” (or something to that effect). Referring to original algorithm as GAIL-log, and the modification as GAIL could be misleading to readers.


3.	The authors claim that SAIL has better training stability, leading to more robust policies. If this is due to the algorithmic contribution of combining AIL and Support-Estimation-IL, then GAIL-log and SAIL-log in Table 2. should show this in the standard deviation numbers. This doesn’t appear to be the case. Also, Figure 4 (Half-Cheetah) has unusually large variance for SAIL-log.


4.	Figure 4 and Table 2 numbers are very different. Take Humanoid for instance. From Figure 4, it seems that SAIL is way better than GAIL. But if you look at Table 2, they both achieve mean-score in excess of 10k. What’s the difference between Table 2. and final performance in Figure 4?


**Experience Assessment:**

I have published one or two papers in this area.

**Review Assessment: Checking Correctness Of Derivations And Theory:**

I assessed the sensibility of the derivations and theory.

**Review Assessment: Checking Correctness Of Experiments:**

I assessed the sensibility of the experiments.

**Review Assessment: Thoroughness In Paper Reading:**

I read the paper thoroughly.

---

> ### Author Response · Authors · 2019-11-11
> **Response to Reviewer 3**
>
> We thank the reviewer for the constructive feedback. We clarify the concerns below.
>
> - On addressing reward bias
> We agree that the MDP raised by the reviewer can’t be fully handled by SAIL unless a smaller discount factor is used. We have updated the paper to discuss this limitation.
>
> We have included a comparison with the absorbing state (AS) technique (Kostrikov et al. 2019) in the Table 5 from Appendix. We also run SAIL combined with AS, since the two methods are not mutually exclusive.
>
> The results are:
> Default env
> GAIL 258.30±28.98 | GAIL+AS  271.46±11.90 | SAIL 262.97±18.11 | SAIL+AS 270.33±15.86
> Modified env (renamed as “Goal-terminal” env in the appendix)
> GAIL -7.16±31.64 | GAIL+AS 110.22±119.25 | SAIL 252.07±67.22 | SAIL+AS 258.30±20.75
>
> As the reviewer suggested, AS improves GAIL significantly in both environments. In particular, in the default environment, GAIL+AS has performance comparable to (or slightly better than) SAIL.
>
> However, we also observe that: 1) SAIL+AS either outperforms (or is comparable to) GAIL+AS, showing that the proposed approach is generally more favorable than GAIL. Also, SAIL+AS has much smaller variance than standard SAIL. 2) In the modified environment, GAIL+AS is unable  to reach the expert’s performance and suffers from a high variance, even when it improves significantly upon GAIL. On the contrary, SAIL and SAIL+AS are on-par with the expert.
>
> - On generalization of SAIL
> To better study the effect of SAIL, we further modified LunarLander to contain no terminal states at all. We refer to this setting as NoTerm. Here each episode ends only after a fixed time limit (1000 steps). In this environment, the absorbing state (AS) method is not applicable. We obtain the following returns (updated Table 1 in the paper):
>
> GAIL 169.73±80.84 | SAIL 256.83±20.99
>
> showing that SAIL is significantly more robust than GAIL also in this setting.
>
> - On GAIL not converging to the same policy in the default and modified environments:
> We observe that in all variants of LunarLander, the policies learned by GAIL, GAIL+AS, SAIL and SAIL+AS  keep oscillating between the partially hovering (i.e. hopping) behavior and the landing behavior during training. The oscillation is due to the stochastic optimization and the finite number of expert demonstrations. GAIL is less consistent in the modified environment than in the default one, which leads to lower performance reported.
>
> In summary, the results on all variants of LunarLander suggests that AS and SAIL both mitigate the implicit reward bias.
>
> - On Fig 3
> We agree that RL optimizes for Q values rather than immediate reward. Fig 3 is only intended as a visualization to support our intuition that the shaping effect form support guidance favors the expert actions more, ultimately contributing to more robust algorithm.
>
> - Clarification on Naming
>  We thank the reviewer for the potential confusion over denoting the methods. We have renamed the bounded variants as GAIL-b and SAIL-b respectively, to avoid confusion.
>
> - Clarification on Mujoco results
>
> Fig 4 and Table 2 report different aspects of the experiment.
>
> For each task and algorithm, we train 5 policies, each initialized by a different random seed. Fig 4 shows the mean performance and standard deviation across the 5 policies. The standard deviation thus shows the sensitivity of different algorithms with respect to the seeds. In contrast, Table 2 reports the performance of the best policy among the 5 for each task and algorithm. The standard deviation in Table 2 shows the robustness/stability of policies with respect to the environment.
>
> To address the specific concerns raised, the large standard deviation for SAIL-log on HalfCheetah was due to 1 out 5 policies converging to a sub-optimal one with performance near  1.7k) Similarly for GAIL, 2 out of 5 policies converged to sub-optimal ones (average performance near 2k), and thus lowering the average value in Fig 4. The best policy from the 5 policies  was reported in Table 2, which achieved over 10k.
>
> -On large variance for SAIL-log, and why SAIL-log may perform worse than GAIL-log
> In Table 2, SAIL-log has larger standard deviations as the product reward in this case is biased towards adversarial discriminator. This is because -Log D is in [0, inf] when r_red is in [0, 1]. In contrast, SAIL receives equal contribution from support guidance and adversarial reward as both components have [0. 1] range. That’s why we proposed and advocate for the use of the bounded reward.

---

### Decision · Program_Chairs · 2019-12-19

**Decision:**

Reject

**Comment:**

The submission proposes a method for adversarial imitation learning that combines two previous approaches - GAIL and RED - by simply multiplying their reward functions. The claim is that this adaptation allows for better learning - both handling reward bias and improving training stability.

The reviewers were divided in their assessment of the paper, criticizing the empirical results and the claims made by the authors. In particular, the primary claims of handling reward bias and reducing variance seem to be not well justified, including results which show that training stability only substantially improves when SAIL-b, which uses reward clipping, is used.

Although the paper is promising, the recommendation is for a reject at this time. The authors are encouraged to clarify their claims and supporting experiments and to validate their method on more challenging domains.